# Acalculous Cholecystitis in a Young Adult with Scrub Typhus: A Case Report and Epidemiology of Scrub Typhus in the Maldives

**DOI:** 10.3390/tropicalmed6040208

**Published:** 2021-12-08

**Authors:** Hisham Ahmed Imad, Aishath Azna Ali, Mariyam Nahuza, Rajan Gurung, Abdulla Ubaid, Aishath Maeesha, Sariu Ali Didi, Rajib Kumar Dey, Abdullah Isneen Hilmy, Aishath Hareera, Ibrahim Afzal, Wasin Matsee, Wang Nguitragool, Emi. E. Nakayama, Tatsuo Shioda

**Affiliations:** 1Mahidol Vivax Research Unit, Faculty of Tropical Medicine, Mahidol University, Bangkok 10400, Thailand; wang.ngu@mahidol.edu; 2Department of Viral Infections, Research Institute for Microbial Diseases, Osaka University, Osaka 565-0871, Japan; emien@biken.osaka-u.ac.jp (E.E.N.); shioda@biken.osaka-u.ac.jp (T.S.); 3Department of Surgery, Indira Gandhi Memorial Hospital, Malé 20002, Maldives; dr.azna@igmh.gov.mv (A.A.A.); dr.mariyamnahuza@igmh.gov.mv (M.N.); dr.rajangurung@igmh.gov.mv (R.G.); dr.ubaid@igmh.gov.mv (A.U.); 4Department of Medicine, Indira Gandhi Memorial Hospital, Malé 20002, Maldives; dr.aishathmaesha@igmh.gov.mv (A.M.); sariualididi@igmh.gov.mv (S.A.D.); dey@igmh.gov.mv (R.K.D.); hlmabd001@myuct.ac.za (A.I.H.); 5Gastrointestinal Unit, Department of Medicine, Groote Schuur Hospital, University of Cape Town, Cape Town 7935, South Africa; 6Health Protection Agency, Ministry of Public Health, Malé 20002, Maldives; hareera@health.gov.mv (A.H.); afzal@health.gov.mv (I.A.); 7Department of Clinical Tropical Medicine, Faculty of Tropical Medicine, Mahidol University, Bangkok 10400, Thailand; wasin.mat@mahidol.edu; 8Department of Molecular Tropical Medicine and Genetics, Faculty of Tropical Medicine, Mahidol University, Bangkok 10400, Thailand

**Keywords:** acute acalculous cholecystitis, *Orientia tsutsgugamushi*, scrub typhus, eschar, Epstein–Barr virus, re-activation, clinical manifestation, Maldives

## Abstract

Scrub typhus is a neglected tropical disease predominantly occurring in Asia. The causative agent is a bacterium transmitted by the larval stage of mites found in rural vegetation in endemic regions. Cases of scrub typhus frequently present as acute undifferentiated febrile illness, and without early diagnosis and treatment, the disease can develop fatal complications. We retrospectively reviewed de-identified data from a 23-year-old woman who presented to an emergency department with complaints of worsening abdominal pain. On presentation, she appeared jaundiced and toxic-looking. Other positive findings on abdominal examination were a positive Murphey’s sign, abdominal guarding and hepatosplenomegaly. Magnetic resonance cholangiopancreatography demonstrated acalculous cholecystitis. Additional findings included eschar on the medial aspect of the left thigh with inguinal regional lymphadenopathy. Further, positive results were obtained for immunoglobulins M and G, confirming scrub typhus. The workup for other infectious causes of acute acalculous cholecystitis (AAC) detected antibodies against human herpesvirus 4 (Epstein–Barr virus), suggesting an alternative cause of AAC. Whether that represented re-activation of the Epstein–Barr virus could not be determined. As other reports have described acute acalculous cholecystitis in adult scrub typhus patients, we recommend doxycycline to treat acute acalculous cholecystitis in endemic regions while awaiting serological confirmation.

## 1. Introduction

Infectious diseases are well-known as a scourge of wars. Some infamous accounts include endemic typhus during the Franko–Russo war, trench fever in World War I, and scrub typhus during World War II [1]. The latter of these, scrub typhus, is caused by the rickettsial species *Orientia tsutsugamushi*, an intra-cytosolic coccobacillus with three main prototype strains (Karp, Gilliam and Kato) that can inflict disease of varying severities in humans [2].

The earliest description of scrub typhus appears to be an account of “shashitsu” in Chinese literature from 313 AD [3]. Centuries later, the word “tsutsugamushi” was used in Japan to describe a febrile illness following insect exposures. Tsutsugamushi was a feverish delirium associated with regional lymphadenopathy, rash and a pathognomonic cutaneous finding of tache noire or black spot, commonly referred to now as eschar [4]. 

The skin lesion of eschar is characterized by central necrosis and surrounding erythema with a halo sign [5]. The necrosis results from acidic enzymes channeled through a stylosome, a funnel created by the vector, gaining access via a skin pore or hair follicle [6]. This triggers a host response comprising infiltration of white blood cells into the dermis, causing the peripheral crusting of the scab.

In the Asia-Pacific theater of World War II, stringent vector control measures were employed to protect deployed troops from the parasitic larval stage of a chigger-mite found amongst the vegetation in endemic regions [7,8]. Nevertheless, this task proved challenging during the pre-antibiotic era and case fatality reached up to 40%. In 1947, the first cases of successful treatment with chloromycetin were reported from the Malay Peninsula [9,10]. 

However, despite the modern availability of surefire antibiotics with bacteriostatic properties that stall the bacterium *O. tsutsugamushi* and allow phagocytes to clear up the infection, the disease still occurs after exposure to the vector. Escalating this problem is the geographic expansion of previously restricted endemic regions, and cases clinically compatible with scrub typhus or showing serological evidence have been reported from Chile, the United Arab Emirates and parts of Africa [11,12,13,14,15,16,17,18].

The Maldives is one endemic region for scrub typhus in Asia [5]. The first description of scrub typhus from the Maldives was reported during World War II [8]. The archipelago has a peculiar historical account comprising a six-decade period with absence of the disease ended by re-emergence with 168 cases, including 10 fatalities [19]. Since that re-emergence in the Maldives, sentinel reporting of scrub typhus has shown occurrences throughout the year, with two periodic peaks observed during festive Eid holiday seasons, which falls during the ninth and twelfth months of the lunar calendar [20]. 

Complications of scrub typhus often involve multiple systems and fatalities are often due to multi-organ dysfunction syndrome [21]. Commonly involved systems include the respiratory, central nervous, cardiovascular, renal, and gastrointestinal systems. However, reports of complications such as pancreatitis, acute cholecystitis, or acalculous cholecystitis in scrub typhus are rare.

First described in 1844 by a surgeon, acute acalculous cholecystitis (AAC) occurs in mainly critically ill geriatric patients. AAC also occurs postoperatively, and in patients requiring medical interventions such as mechanical ventilation or prolonged parenteral nutrition and fasting [22,23]. This pathology accounts for 10% of all cases of acute cholecystitis and is associated with high morbidity and mortality rates [24,25]. The pathophysiology involves inflammation of the gallbladder causing bile stasis and disruption of circulation causing ischemia and necrosis or rupture of the gallbladder, which then involves the peritoneum. In such cases, AAC is associated with infectious etiologies, mainly involving Gram-negative Enterobacteriaceae or Gram-positive spore-forming bacilli [26]. Interestingly, some other pathogens such as Epstein–Barr virus and *O. tsutsugamushi* have been described to be associated with AAC in otherwise healthy adults [27,28,29,30,31,32,33,34,35,36,37,38]. Here, we report a case of acute acalculous cholecystitis in a young healthy adult from the Maldives.

## 2. Materials and Methods

The case we describe here presented to Indira Gandhi Memorial Hospital (IGMH) in Malé, Republic of Maldives, in July 2021. De-identified clinical and laboratory data during the hospitalization were reviewed using in-patient medical charts. Diagnostics for *O. tsutsugamushi* included the “Scrub Typhus Detect™” immunoglobulin (Ig)M/IgG enzyme-linked immunosorbent assay (InBios International, Seattle, WA, USA). Diagnostics for Epstein–Barr virus included VIDAS®EBV which detected Epstein–Barr viral capsid antigen (EBV-VCA) IgM/IgG, early antigen (EBV-EA) IgG, and nuclear antigen (EBV-NA) IgG by chemiluminescence assays (bioMérieux Italia, Florence, Italy). Hemoculture was performed using an automated culture system (bioMérieux, Durham, NC, USA). We also obtained epidemiological surveillance data for the preceding nine years (2012–2021) on scrub typhus in the Maldives from the Health Protection Agency at the Ministry of Health.

## 3. Case Report

We present the case of a previously healthy 23-year-old woman who presented to the emergency department of IGMH with worsening abdominal pain and fatigue (Appendix A). The presenting complaint was associated with a self-reported 9-day history of high-grade fever accompanied by chills, and displaying a diurnal pattern. Four days before presentation, the patient had consulted a primary-care physician and had been prescribed amoxicillin and clavulanic acid for tonsillopharyngitis. Additional history included several episodes of vomiting with the onset of colicky, non-radiating abdominal pain of severe intensity, without any apparent aggravating or relieving factors. The patient had traveled from her home island of Gadhdhoo, in a southern atoll in the Maldives, to the capital city of Malé to visit friends and relatives a couple of days before falling ill, as illustrated in Appendix A. The annual descriptive epidemiology of scrub typhus in the Maldives is presented in Appendix A.

On examination, the patient was conscious and coherent, but appeared dehydrated, jaundiced, and toxic-looking. Oral temperature was 39 °C, and heart rate was 110 beats/min with a mean arterial blood pressure of 73 mmHg. Both tonsils were enlarged with evidence of pharyngitis and palpable cervical lymphadenopathy. Evaluations of the cardiovascular, respiratory, and central nervous systems found no abnormalities. Other abdominal examinations revealed a positive Murphy’s sign and guarding on palpation of the abdomen. The results of laboratory investigations on presentation are shown in Table 1.

The initial hematological profile on presentation showed lymphocytosis and increased monocytes, although leukocytes, neutrophils, and platelets remained within normal ranges. Hyperbilirubinemia and elevated transaminases were identified from liver function tests. Further, hyperferritinemia and elevated levels of C-reactive protein were identified. Ultrasonography confirmed Murphey’s sign from the probe test and demonstrated edema of the gallbladder wall. Computed tomography (CT) of the abdomen revealed hepatosplenomegaly. Magnetic resonance cholangiopancreatography subsequently confirmed acalculous cholecystitis, as shown in Figure 1.

The patient was admitted with parenteral fluids to correct dehydration, broad-spectrum antibiotics (cefotaxime at 2 g/day, metronidazole at 1500 mg/day), and parenteral analgesics. During nursing care, eschar was noticed on the medial aspect of the left thigh (Appendix A. Left regional inguinal lymphadenopathy was also detected. Doxycycline was started at a loading stat dose of 200 mg followed by 200 mg/day in two divided doses for seven days. Fever cleared within 52 h of beginning doxycycline (Appendix A. Symptoms of abdominal pain gradually improved, and at the time of discharge from hospital, she had residual signs of fatigue.

## 4. Discussion

In the Maldives, scrub typhus was simply forgotten until its sudden re-emergence in 2002 [5,19,39]. Since decades of sentinel data reflect the existence of *O. tsutsugamushi* throughout the archipelago, reports of scrub typhus from Faafu, Vaavu, and Meemu atolls are recent. We loosely associate this peculiar finding with urbanization, which could have exposed individuals to areas of vegetation where the vector is found [5].

The incubation of scrub typhus is 6–21 days [21]. Consequently, when obtaining a travel history, a minimum of three weeks is necessary to consider scrub typhus among the differential diagnoses. Clinical manifestations of scrub typhus commonly appear first as undifferentiated febrile illness. During the early phase of disease, systemic symptoms are predominantly present without localizing foci of infection. Nevertheless, our patient presented with intermittent fever and she had complaints of sore throat and documented findings of tonsillopharyngitis from her initial visit to the primary-care physician. Similar manifestations had previously been described in scrub typhus patients, where misdiagnosis unfortunately led to complications from scrub typhus [40]. 

Gastrointestinal (GI) symptoms occur in scrub typhus, but less frequently compared to systemic symptoms [41]. GI symptoms were reported in 20–76% of scrub typhus cases, with varying severity [42,43]. As described in one cohort, the most common GI symptom was nausea, followed by abdominal pain, vomiting, and diarrhea [41]. The patient in the present case developed nausea during the third day after symptoms onset and experienced several episodes of vomiting with bile contents, up until the abdominal pain started. A correlation between endoscopic findings suggestive of gastrointestinal vasculitis and disease severity has previously been demonstrated [43]. *O. tsutsugamushi* shows tropism to several host cells, including endothelial cells with the pathogenicity of endotheliitis [44]. Dissemination occurs within macrophages, and the bacterium has previously been detected in several organs, including in the human gallbladder [45]. 

Other studies have previously described hepatosplenomegaly, gallbladder or periportal edema and abdominal lymphadenopathy as GI findings visualized in patients with scrub typhus [46]. In the same cohort, splenomegaly was detected in 15 of 19 cases and 16% developed splenic infarcts due to acute enlargement of the spleen. The patient in our case presented also exhibited massive splenomegaly but, fortunately, no splenic infarcts developed. In addition, the liver was enlarged. Animal studies have demonstrated hepatomegaly resulting from periportal inflammation [47]. Similarly, thickening of the gallbladder wall can result from perivasculitis or vasculitis [32]. The hyperbilirubinemia observed in the presented case was likely due to vasculitis and hepatitis from *O. tsutsugamushi* as previously described [31,48]. Nevertheless, as there was no biopsy performed, we could not demonstrate the histopathological changes compatible with cholestatic hepatitis. Presentation with acute abdomen in cases of scrub typhus is rare, but does occur as in this case, and has on occasion resulted in explorative surgeries [49].

The patient in this case showed the triad of fever, right upper quadrant pain, and clinical jaundice, suggestive of acute cholecystitis. Nevertheless, some hematological features such as leukocytosis were absent from the hematological profile. In addition, no risk factors for cholecystitis were identified, such as alcohol consumption or evidence of cholelithiasis. With the identification of the eschar during nursing care, scrub typhus was suspected and doxycycline was started, and the fever cleared within two days. Other antibiotics commonly used to treat scrub typhus are chloramphenicol and azithromycin, with alternative drugs such as rifampin as a second-line agent in tuberculosis endemic regions [50]. The patient was discharged from hospital five days later, once the abdominal pain had completely subsided. We were unable to obtain follow-up images showing resolution of gallbladder wall edema or a clinical specimen for molecular confirmation of *O. tsutsugam ushi*, due to the retrospective nature of this report.

In tropical regions, coinfections with other tropical diseases are often observed, obfuscating the clinical characteristics of diseases [51]. This provides a major impediment to correct clinical judgment and often results in a poor clinical trajectory with missed diagnoses [52]. It has been previously reported that 15% of scrub typhus cases had coinfection with other diseases [53]. Identifying coinfections is an important challenge for clinicians, as some are treatable diseases such as influenza, malaria, leptospirosis, typhoid or melioidosis [54,55,56,57,58,59,60,61,62,63]. Others have described severe manifestation in scrub typhus co-infected with dengue and other viral infections [64,65,66,67,68].

Interestingly, EBV was also detected from serology, and these results should be interpreted carefully. The patients showed several features consistent with primary mononucleosis, including tonsillopharyngitis, adenopathy, hepatosplenomegaly, and lymphocytosis. In addition, no eruption of rash was identified in this case, as previously described [69]. We could not determine the percentage increase in atypical lymphocytes, which would have helped distinguish primary mononucleosis from reactivation [70]. Reactivation is frequently overserved with individuals with primary or secondary immunodeficiency states [71]. Further, reactivation of EBV from other intracellular bacteria and viruses has previously been described [72,73]. Nevertheless, others have also reported mononucleosis-like features in scrub typhus [74]. As mentioned in the Introduction, EBV infection can also complicate and manifest as acute cholecystitis without lithiasis. During the lytic phase, the virus can replicate in tonsillar tissues and hepatocytes and manifest with splenomegaly. We wondered whether *O. tsutsugamushi* could trigger the activation of latent EBV and whether reactivation of B-lymphotrophic oncogenic viruses such as EBV pose a risk for the development of malignancies [75]. 

Scrub typhus is a neglected tropical disease and a public health concern. Increasing public awareness regarding prevention and educating clinicians beginning work in endemic regions about scrub typhus, among other common tropical diseases, could help detect cases, start treatment in a timely manner, and prevent complications and death [76].


In conclusion, scrub typhus can complicate and manifest as acute acalculous cholecystitis in young adults. Therefore, we recommend using doxycycline empirically in cases presenting from regions endemic for scrub typhus, providing no contraindications are present.

## Figures and Tables

**Figure 1 tropicalmed-06-00208-f001:**
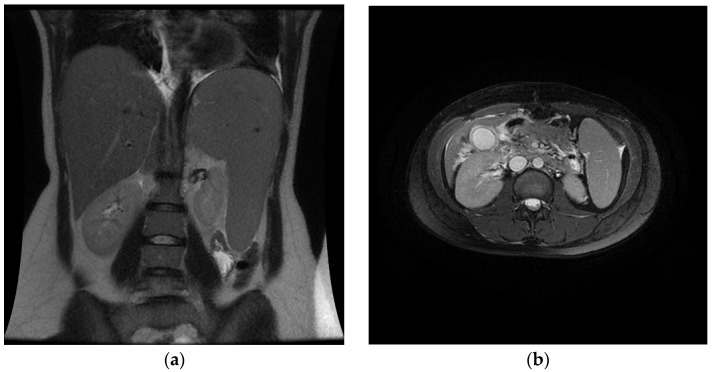
Imaging of the abdomen during hospitalization. (**a**) Coronal-view CT of the abdomen shows hepatosplenomegaly. (**b**) Axial-view CT of the abdomen demonstrates cholecystitis with enhancement of the gallbladder wall. (**c**) Longitudinal ultrasonography depicts an edematous gallbladder wall (wall thickness > 4 mm) without lithiasis. (**d**) Magnetic resonance cholangiopancreatography reveals a ballooned gallbladder without any ductal obstruction, but with signs of inflammation consistent with acalculous cholecystitis.

**Table 1 tropicalmed-06-00208-t001:** Laboratory Results at Presentation and Before Discharge.

Day of Illness (Days)	9	14
Leukocyte/µL	6650	8030
Neutrophils/µL	4788	4496
Lymphocytes/µL	1349	2328
Monocytes/µL	435	1003
Eosinophils/µL	0	24
Basophils/µL	0	0
Platelets/µL	269,000	545,000
Hemoglobin (g/dL)	10.6	7.8
Hematocrit (%)	32.9	24.6
Total Bilirubin (mg/dL)	4.7	3.5
Direct Bilirubin (mg/dL)	3.5	2.6
Total protein (g/dL)	6.9	6.8
Albumin (g/dL)	3.4	2.8
Alkaline phosphatase (IU/L)	288	250
Aspartate aminotransferase (IU/L)	140	76
Alanine aminotransferase (IU/L)	157	93
Creatinine (mg/dL)	0.8	0.6
Urea (mg/dL)	10	6.4
CRP (mg/dL)	7.7	2.8
Sodium (mmol/L)	131	134
Potassium (mmol/L)	4.5	4.3
Ferritin (ng/mL)	1299.3	
LDH (IU/L)	726	
EBV-VCA IgM (IU/mL)	63.6	
EBV-VCA IgG (IU/mL)	89.9	
EBV-EA IgG (IU/mL)	16.1	
EBV-NA IgG (IU/mL)	574.0	
Blood culture		no growth
*Orientia tsutsugamushi* IgM	positive	
*Orientia tsutsugamushi* IgG	positive	

CRP: C-reactive protein, LDH: lactate dehydrogenase, EBV-VCA: Epstein–Barr virus viral capsid antigen, EBV-EA; Epstein–Barr virus early antigen, EBV-NA: Epstein–Barr virus nuclear antigen, IgM: immunoglobulin M, IgG: immunoglobulin G.

## Data Availability

The data presented in this study are available on request from the corresponding author. The data are not publicly available to ensure the privacy of the study participant.

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
