# Peer review of "Acalculous Cholecystitis in a Young Adult with Scrub Typhus: A Case Report and Epidemiology of Scrub Typhus in the Maldives"

_tropicalmed, 2021, doi:10.3390/tropicalmed6040208_

Round 1

Reviewer 1 Report

Imad et al described a case of Scrub typhus in a young woman presented with prolonged fever, eschar and acalculus cholecystitis that was treated conservatively with antibiotics. The article is generally well written and interesting. 

I have several comments. 

The most important is the fact that the case is presented as a dual infection of typhus and EBV infection. I dont find evidences for this. The prolonged fever, pharyngitis, hepatosplenomegaly and liver enzymes abnormality all could be assigned to typhus. Scrub typhus presenting with tonsillitis was also described by the authors. The main difficulty to diagnose acute EBV in this case is the presence of EBNA antibodies, which denotes that at least several weeks have passed since acute EBV infection. It may be a reactivation of EBV, but without significant viremia, it is difficult to diagnose this as well. False positive IgM could be very common, especially during another infectious diseases causing vasculitis. Hence, in my opinion, the title of the article should not include EBV as a coinfection and the text should be appropriately amended. 

Other comments:

  1. Introduction - very interesting overview on the disease background and history. There is no much need to repeat and elborate on this in the discussion, and data from there could be added to the introduction.
  2. Intro - when are the holydays in the Maldives? (line 80). 
  3. lines 81-88 - if co-infection is to be less emphasized, this section may be more appropriate in the discussion. There are more papers discussing co infections of Rickettsias with for exapmle typhoid fever or dengue fever. 
  4. case report - were blood cultures taken? were they negative? is it costumed in this facility to use PCR for diagnosis of Scrub typhus? from the blood? form the scub? Was it considered? If not, the authors might want to elaborate on this subject in this case and generally. 
  5. Discussion - the patient had a diurnal pattern of fever, I wonder how did the authors get access to such accurate temperature measurments, while the was not hospitalized. 
  6. Discussion line 261 - I believe that jaundice is not part of the classical presentation of this disease, as opposed with cholangitis. This may implicate that the hepatitis and jaundice were not the consequences of the gallbladder inflammation, but the result of the vasulitis and hepatitis caused by Orientia. If the authors agree, than the discussion should be directed along this line of thought. 
  7. Supplementary : fig S5 - I do not see the green shaded area. 

Author Response

The most important is the fact that the case is presented as a dual infection of typhus and EBV infection. I don’t find evidences for this. The prolonged fever, pharyngitis, hepatosplenomegaly and liver enzymes abnormality all could be assigned to typhus. Scrub typhus presenting with tonsillitis was also described by the authors. The main difficulty to diagnose acute EBV in this case is the presence of EBNA antibodies, which denotes that at least several weeks have passed since acute EBV infection. It may be a reactivation of EBV, but without significant viremia, it is difficult to diagnose this as well. False positive IgM could be very common, especially during another infectious disease causing vasculitis. Hence, in my opinion, the title of the article should not include EBV as a coinfection and the text should be appropriately amended. 

Thank you for your valuable time in reviewing our manuscript. We greatly appreciate the valuable comments forwarded to improve our manuscript. In the manuscript, our intention is to describe the clinical course of a young adult complicated with acute cholecystitis from scrub typhus.

We agree that it would be difficult to prove the re-activation of EBV in the presented case, mainly due to several limitations in this report. First, we could not include EBV viral loads as serology was the only assay used for routine diagnostics for EBV in atypical cases at the hospital. Further, there were no subsequent serological assays to provide additional information to demonstrate a false-positive IgM result.

As suggested by the reviewer, we have revised the title to “Acalculous cholecystitis in a young adult with scrub typhus: a case report and epidemiology of scrub typhus in the Maldives”

We further tone done on the interpretation of the EBV heterophile antibodies. In the revised manuscript, we have brought the following changes.

  • Lines 36 to 39: we have rephrased the sentences in the Abstract to “The workup for other infectious causes of acute acalculous cholecystitis (AAC) detected antibodies against human herpesvirus 4 (Esptein-Barr virus), suggesting an alternative cause of AAC. Whether that represented re-activation of the Epstein-Barr virus could not be determined.”
  • Lines 286-287: we have rephrased the sentence in the Discussion to “Interestingly, EBV was also detected from serology, and these results should be interpreted carefully.

Other comments:

  1. Introduction - very interesting overview on the disease background and history. There is no much need to repeat and elborate on this in the discussion, and data from there could be added to the introduction.

We are pleased to read the kind words regarding the overview of the disease. As suggested by the reviewer we have revised the Discussion and added the information to the Introduction in the revised manuscript (Lines 76-83)

  1. Intro - when are the holydays in the Maldives? (line 80). 

The long holidays in the Maldives are observed during the Eid festivals. These are week-long holidays where many of the population visit back to their native islands on vacation. Picnic trips within the atoll expose individuals to chigger-infested islands. The onset of illness beings upon returning from their vacation.

Lines 87 to 88: we have revised the sentence in the revised manuscript to “Since that re-emergence in the Maldives, sentinel reporting of scrub typhus has shown occurrences throughout the year, with two periodic peaks observed during festive Eid holiday seasons, which falls during the ninth and twelfth months of the lunar calendar.”

  1. lines 81-88 - if co-infection is to be less emphasized, this section may be more appropriate in the discussion. There are more papers discussing co infections of Rickettsias with for exapmle typhoid fever or dengue fever. 

As suggested by the reviewer, we have emphasized about co-infections in Rickettsiosis in the Discussion. In the revised manuscript, we have added the following paragraph in Lines 278-285.

In tropical regions, coinfections with other tropical diseases are often observed, obfuscating the clinical characteristics of diseases. This provides a major impediment to correct clinical judgment and often results in a poor clinical trajectory with missed diagnoses. It has been previously reported that 15% of scrub typhus cases had coinfection with other diseases. Identifying coinfections is an important challenge for clinicians, as some are treatable diseases such as influenza, malaria, leptospirosis, typhoid or melioidosis. Others have described severe manifestation in scrub typhus co-infected with dengue and other viral infections.

  1. case report - were blood cultures taken? were they negative? is it costumed in this facility to .use PCR for diagnosis of Scrub typhus? from the blood? form the scub? Was it considered? If not, the authors might want to elaborate on this subject in this case and generally. 

Blood cultures were taken and they did not yield any bacteria. We have included these results in Table 1 and also mentioned regarding this as one of the limitations (Line 275-277). Unfortunately, the hospital is not resourced enough for molecular confirmation of O. tsutsugamushi. Further due to the COVID-19 pandemic we faced some restriction on importing clinical specimen (scab) to labs in Thailand and Japan for molecular confirmation O. tsutsugamushi.

  1. Discussion - the patient had a diurnal pattern of fever, I wonder how did the authors get access to such accurate temperature measurements, while the was not hospitalized. 

Thank you for this question and please allow us to explain this point. We had access to de-identified digitized information collated from the medical chart of the presented case. We are grateful to the competent and excellent clinical clerkship of some the co-authors who had included substantial detailed history in chronological order as the presenting history of illness. Especially when the case had worsening abdominal pain nine days after developing fever and other symptoms. From our retrospective review of the available data of the presented case, we gathered that the patient had a diurnal pattern prior to presentation.

In lines 127-128 we have revised the sentence in the revised manuscript to “The presenting complaint was associated with a self-reported 9-day history of high-grade fever accompanied by chills, and displaying a diurnal pattern.”

Further in the supplementary figure, we have added the following foot note to Figure S1,

 “Temperature depicted is not the actual reading, but graphically presented to demonstrate the fever pattern described by the patient.”

  1. Discussion line 261 - I believe that jaundice is not part of the classical presentation of this disease, as opposed with cholangitis. This may implicate that the hepatitis and jaundice were not the consequences of the gallbladder inflammation, but the result of the vasulitis and hepatitis caused by Orientia. If the authors agree, than the discussion should be directed along this line of thought. 

We agree with the reviewer’s suggestion. In the revised manuscript, we have addition the following sentences (Lines 259-263)

“The hyperbilirubinemia observed in the presented case was likely due to vasculitis and hepatitis from O. tsutsugamushi as previously described. Nevertheless, as there was no biopsy performed, we could not demonstrate the histopathological changes compatible with cholestatic hepatitis.”

  1. Supplementary : Figure S5 - I do not see the green shaded area. 

The supplementary Figure S5 includes the shaded area as shown below.

Reviewer 2 Report

It is an interesting Case Report.
However, there are many issues that must be addressed.

1) The Introduction section is too extended and a proportion of the information there is mentioned in the Discussion section

2) Table 1:
a) Is the value of lymphocytes correct? Units?
b) Substitute word "hemoculture" with "blood culture"

3) Please enrich the conversation regarding the serological tests for EBV  explaining further why do you think that it might be reactivation of the virus. When do we usually observe reactivation of the EBV?

4) In the Discussion section, please reduce the length of the historical flashback and focus to explaining the findings of your case (e.g., how do you explain the observed hyperbilirubinemia?)

5) The dosage of doxycycline in your case is somewhat unusual. Could you provide more information regarding this, other dosage regimens, and alternative antibiotics?

Author Response

Comments and Suggestions for Authors

It is an interesting Case Report.
However, there are many issues that must be addressed.

Thank you for your valuable time reviewing our manuscript, and we appreciate the suggestions to help improve our manuscript.

1) The Introduction section is too extended and a proportion of the information there is mentioned in the Discussion section

We wanted to emphasize the epidemiology of scrub typhus in the Maldives in the Discussion. Nevertheless, to avoid repetitions, we have revised the Discussion and move the information to the Introduction (Lines 76-83). Therefore, we kindly request you to consider the Introduction as it stands without further shortening it.

2) Table 1:
a) Is the value of lymphocytes correct? Units?
b) Substitute word "hemoculture" with "blood culture"

Thank you for bringing this error to our attention. In the revised manuscript Table 1, we have added the unit (µL). Further in the table, we have also substituted the word hemoculture with blood culture.

3) Please enrich the conversation regarding the serological tests for EBV explaining further why do you think that it might be reactivation of the virus. When do we usually observe reactivation of the EBV?

Thank you for this suggestion. In the revised manuscript, we have discussed to caution regarding the interpretation of serological result of EBV. We have rephrased the sentence in Line 286-287 to the following:

Interestingly, EBV was also detected from serology, and these results should be interpreted carefully.”

Further in the Discussion, we have added the following sentence regarding reactivation in the revised manuscript (Line 292-294)

Reactivation is frequently overserved with individuals with primary or secondary immunodeficiency states. Further, reactivation of EBV from other intracellular bacteria and viruses has previously been described.

4) In the Discussion section, please reduce the length of the historical flashback and focus to explaining the findings of your case (e.g., how do you explain the observed hyperbilirubinemia?).

Thank you for this suggestion. In the revised manuscript, we have moved the paragraph in the Discussion to the Introduction (Lines 76-83)

Further we have included the following sentence in the Discussion (Lines 259-263)

“The hyperbilirubinemia observed in the presented case was likely due to vasculitis and hepatitis from O. tsutsugamushi as previously described. Nevertheless, as there was no biopsy performed, we could not demonstrate the histopathological changes compatible with cholestatic hepatitis.”

5) The dosage of doxycycline in your case is somewhat unusual. Could you provide more information regarding this, other dosage regimens, and alternative antibiotics?

As suggested we have added the following sentence to the Discussion in Lines 271-274

Other antibiotics commonly used to treat scrub typhus are chloramphenicol and azithromycin, with alternative drugs such as rifampin as a second-line agent in tuberculosis endemic regions.

The dosage of doxycycline administered to case presented was 2 capsules of 100mg doxycycline stat and 1 capsule of 100mg doxycycline twice daily.

We have revised the sentence (Lines 203-204) to following sentence in the revised manuscript.

“Doxycycline was started at a loading stat dose of 200 mg followed by 200 mg/day in two divided doses for seven days.”

Round 2

Reviewer 1 Report

The authors have made the required adaptation to the text. 

I still cannot find the green shadow in figure 5S

Author Response

The supplementary file uploaded contains the green shaded area.

I have attached the word file with the image for your reference 

Reviewer 2 Report

In table 1, the sum of lymphocytes and neutrophils (without even calculate monocytes, basophils and eosinophils) is > of the total leukocytes.
In the previous review round i have asked you if the value of lymphocytes is correct for that reason.

Author Response

We have corrected the differential counts in Table 1, in the revised manuscript.